# Bioinformatic Analysis of a Set of 14 Temperate Bacteriophages Isolated from *Staphylococcus aureus* Strains Highlights Their Massive Genetic Diversity

Cristian A. Suárez,[a] Soledad T. Carrasco,[a] Facundo N. A. Brandolisio,[a] Virginia Abatangelo,[a] Carina A. Boncompain,[a] Natalia Peresutti-Bacci,[a] Héctor R. Morbidoni[a]

[a]Laboratorio de Microbiología Molecular, Facultad de Ciencias Médicas, Universidad Nacional de Rosario, Rosario, Argentina

**ABSTRACT** Epidemiology and virulence studies of *Staphylococcus aureus* showed that temperate bacteriophages are one of the most powerful drivers for its evolution not only because of their abundance but also because of the richness of their genetic payload. Here, we report the isolation, genome sequencing, and bioinformatic analysis of 14 bacteriophages induced from lysogenic *S. aureus* strains from human or veterinary (cattle) origin. The bacteriophages belonged to the *Siphoviridae* family; were of similar genome size (40 to 45 kbp); and fell into clusters B2, B3, B5, and B7 according to a recent clustering proposal. One of the phages, namely, vB_SauS_308, was the most unusual one, belonging to the sparsely populated subcluster B7 but showing differences in protein family contents compared with the rest of the members. This phage contains a type I endolysin (one catalytic domain and noncanonical cell wall domain [CBD]) and a host recognition module lacking receptor binding protein, cell wall hydrolase, and tail fiber proteins. This phage also lacked virulence genes, which is opposite to what has been reported for subcluster B6 and B7 members. None of six phages, taken as representatives of each of the four subclusters, showed activity on coagulase-negative staphylococci (excepted for two *Staphylococcus hominis* strains in which propagation and a very slow adsorption rate were observed) nor transducing ability. Immunity tests on *S. aureus* RN4220 lysogens with each of these phages showed no cross immunity.

**IMPORTANCE** To the best of our knowledge, this set of sequenced bacteriophages is the largest one in South America. Our report describes for the first time the utilization of MultiTwin software to analyze the relationship between phage protein families. Notwithstanding the fact that most of the genetic information obtained correlated with recently published information, due to their geographical origin, the reported analysis adds up to and confirms currently available knowledge of *Staphylococcus aureus* temperate bacteriophages in terms of phylogeny and role in host evolution.

**KEYWORDS** bacteriophages, *Staphylococcus aureus*, genomic evolution

Bacteriophages (phages) capable of infecting *Staphylococcus aureus* have been known and studied as early as 1930 (1) and were used in a simple method of bacterial typing (2, 3). However, studies on their role in the formation and evolution of *S. aureus* pathogenesis started in the mid-1960s through the pioneering work of Novick (4), who described a cryptic, high-frequency transducing phage (1967). Further studies on temperate phages helped construct the tools for the molecular manipulation of *S. aureus*, including *Escherichia coli-S. aureus* shuttle plasmids (5). However, one of the most fascinating roles of staphylococcal temperate phages is its involvement in the mobilization of *S. aureus* pathogenicity islands (SaPIs), chromosomal fragments encoding superantigens that can hijack virion particles of induced helper prophages, which are encapsidated and thus transduced at a high frequency (6, 7). Through the studies of this "parasitic" relationship, it was demonstrated that SaPIs can

Address correspondence to Héctor R. Morbidoni, morbiatny@yahoo.com.

The authors declare no conflict of interest.

**TABLE 1** List of *S. aureus* phages isolated throughout this work

| Phages[a] | Source | No. of phages obtained/ no. of samples analyzed | Method of collection | Geographic location |
|---|---|---|---|---|
| 275, 277c, **277g**, 277o, 279, 280, 285g, 285o, **287**, 288, 296, 300, 302, **308**, 312, 313, 314, 318, **320**, **321c**, 321m, 321g, **690**, **713**, 714, 715c, 717, 750, **760**, **775**, 776, **832**, 836, 837, 839, 841, 862 | Human isolates from hands of healthy carriers (meat retail market) | 37/75 | Induction with mitomycin C | La Plata, Buenos Aires, Argentina |
| I72, **I73**, I74, I9 | Cow isolates (mastitis) | 4/23 | Induction with mitomycin C | Rafaela, Santa Fe, Argentina |
| **Mh_1**, Mh_2, Mh_3, **Mh_4**, Mh_5, Mh_8, Mh_10, Mh_11, Mh_14, **Mh_15** | Clinical isolates (blood cultures, soft tissue infections) | 10/18 | Induction with mitomycin C | Rosario, Santa Fe, Argentina |
| Mat_A, Mat_B, Mat_C, Mat_CB, Mat_D, Mat_E, Mat_F, Mat_G, Mat_H, Mat_I, Mat_J, Mat_K, Mat_L, Mat_N, Mat_T, Mat_Y, Mat_W, Mat_X, Mat_Z, Mat_13, Mat_25, Mat_29, Mat_33 | Nasal swabs from health care workers | 23/320 | Spontaneous release | Rosario, Santa Fe, Argentina |

[a]Phages analyzed in detail are shown in bold.

modulate the late gene expression of temperate helper phages (8) and, conversely, that proteins of helper phages can derepress SaPI expression through "moonlight" activities (9). Thus, the mobilization of SaPIs in helper phage capsids became a powerful way to transfer genes intra- and intergenus conferring advantages in terms of adapting to the host and occupying different environmental niches through the expression of superantigens causing different human pathologies, such as the staphylococcal toxic shock syndrome (10). Taking into consideration that most of the phages able to infect *S. aureus* are temperate and that any strain could contain up to 4 of those phages, it is important to study the phylogenetic relationships between them. In this field, a recent report by Oliveira et al. (11) is the first comprehensive attempt to analyze and cluster staphylococcal phages; of 205 genome sequences deposited in public databases, 132 belonged to temperate phages of the *Siphoviridae* family. With the aim of expanding the studies on the genetic diversity of temperate staphylococcal phages, we isolated 74 temperate phages of *S. aureus* strains of human and cattle origin; we report here the genomic sequence and bioinformatics analysis of 14 of those phages.

## RESULTS

**Isolation of bacteriophages from lysogenic *S. aureus* strains.** As part of an effort to isolate temperate bacteriophages residing in the *S. aureus* chromosome, we assembled a set of strains from human and veterinary (mastitis) clinical strains from third parties (see Acknowledgments for details) as well as from healthy volunteers. The induction by mitomycin C treatment of 98 *S. aureus* strains yielded a large number of lysis plaques of which we kept only 1 from each strain; thus, each phage was named after the strain that contained it (Table 1). Each lysis plaque was propagated three times through plating on RN4220 to ensure purity; the plaque morphologies varied from clear to turbid and, in all cases, were pinpoint sized. The genome size of this set of phages was determined by pulsed-field gel electrophoresis (PFGE), revealing sizes ranging from 40 kbp to 50 kbp (data not shown). Here, we report the sequencing and bioinformatics analysis of phages vB_SauS_I73, 277g, 287, 308, 312c, 320, 690, 713, 775, 760, 832, Mh1, Mh4, and Mh15; the remaining phage genome analysis will be reported elsewhere (C.A. Boncompain, unpublished data).

**Sequencing and annotation of the phage genomes.** Genome sequences of the 14 temperate phages were assembled with the A5 pipeline and annotated using the DNAMaster program applying BLASTP, Glimmer, and GeneMarkS software, followed by visual inspection and manual curation. The genomes had an average G+C content of 34.4% (ranging from 33.27% to 35.35%) and a coding DNA sequence (CDS) content between 62 and 75 open reading frames (ORFs) (see Table S1 in the supplemental material). For the sake of comparison, the left end of the genomes was set at the integrase gene. The annotation of the

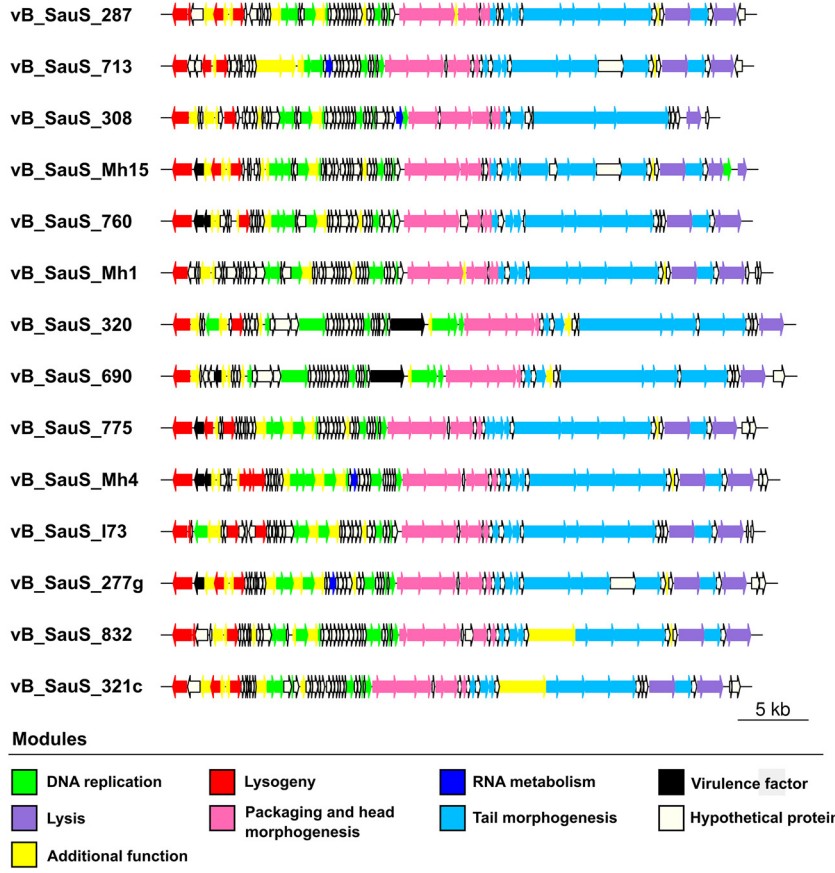

**Modules**

| | | | |
|---|---|---|---|
| 🟩 DNA replication | 🟥 Lysogeny | 🟦 RNA metabolism | ⬛ Virulence factor |
| 🟪 Lysis | 🩷 Packaging and head morphogenesis | 🔵 Tail morphogenesis | ⬜ Hypothetical protein |
| 🟨 Additional function | | | |

**FIG 1** Genomic organization of phages isolated during this work. ORFs were colored according to the assigned modules.

genomes showed from left to right the integration/excision, DNA replication, morphogenesis and packaging, and lysis modules with most of the genes transcribed rightward (Fig. 1). A large number of genes (roughly 45% to 50%) encoding proteins of unknown functions are present through the genomes but concentrated mainly in the DNA replication module (Fig. 1). The predicted proteins are listed in Table S2 in the supplemental material; of note, very few possible toxin or toxin-related genes, such as a toxin-antitoxin system of the MazF type, the virulence-associated protein E, and ImmA/IrrE endopeptidase, are present in our phages (12). Interestingly, phage vB_SauS_308 does not contain genes for any recognizable virulence factor.

**Comparative genomics and phylogeny. (i) Taxonomic classification.** All the isolated phages belong to the order *Caudovirales* and family *Siphoviridae*; in order to classify phages at the genus level, we used vConTACT2 from Cyverse (13). The analysis generated 354 protein clusters; our phages could be classified in four clusters (0_0, 3_0, 9_0, and 9_1). Phages vB_SauS_277g, vB_SauS_321c, vB_SauS_713, vB_SauS_775, vB_SauS_832, and vB_SauS_Mh4 belong to the cluster 9_0; phages vB_SauS_287, vB_SauS_760, vB_SauS_I73, vB_SauS_Mh1, and vB_SauS_Mh15 belong to cluster 9_1; phages vB_SauS_320 and vB_SauS_690 belong to cluster 0_0; and the phage vB_SauS_308 belongs to the cluster 3_0. Phages in clusters 9_0 and 9_1 are members of the new Azeredovirinae subfamily, with those of cluster 9_0 and cluster 9_1 belonging to the genus Dubowvirus and genus Phietavirus, respectively. Moreover, phages of cluster 3_0 and cluster 0_0 belong to the genera Peeveelvirus and Triavirus, respectively.

**(ii) Comparative genomics.** First, 173 genomes of *S. aureus* phages belonging to the *Siphoviridae* family were retrieved from a public database (NCBI). A comparative

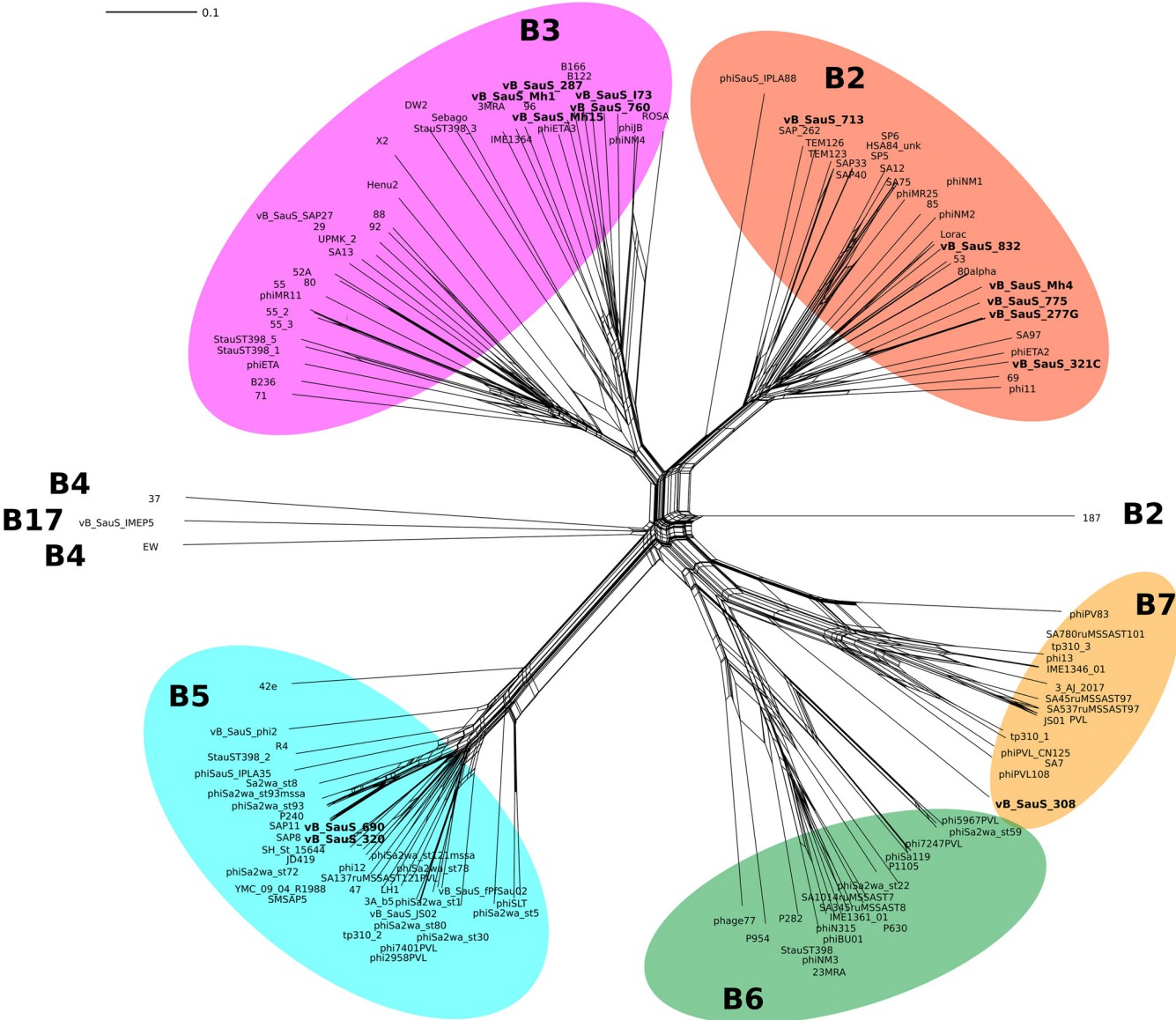

**FIG 2** Phylogenetic network of the *Siphoviridae* phages of *S. aureus*. An all-against-all genome comparison was performed using fragmented alignment by Gegenees (61); afterward, the network was calculated by Splitstree 4.0 using the Neighbor-Net algorithm (62). The majority of phages belong to the subclusters B2, B3, B5, B6, and B7 in accordance with Oliveira et al. (11). Phages described in this work are highlighted in bold.

whole-genome analysis of those phages, in addition to ours, was performed using Gegenees, and the phylogenetic network was obtained using Splitstree 4.0. By means of a Neighbor-Net analysis, four main clusters of phages were identified (Fig. 2). These clusters matched those reported by Oliveira et al. (11); thus, our phages grouped in subclusters B2, B3, B5, or B7. There is a correlation between this grouping and that obtained previously by vConTACT2; thus, phages in subcluster B2 belong to the genus Dubowvirus and phages of subcluster B3 belong to Phietavirus, while those of subcluster B5 and B7 belong to genera Triavirus and Peeveelvirus, respectively.

Comparative genomics was performed using MultiTwin with the purpose of analyzing genes shared by different phages. We retrieved genomes of *Siphoviridae* phages from different hosts from the NCBI database, adding 15 phages from *S. epidermidis*, 12 from *Staphylococcus pseudintermedius*, 3 from *Staphylococcus haemolyticus*, 2 from *Staphylococcus capitis*, 2 from *Staphylococcus hominis*, 2 from *Staphylococcus sciuri*, 1 from *Staphylococcus saprophyticus*, and 1 from *Staphylococcus warneri*. The database we assembled contained 11,800 proteins that

**TABLE 2** MultiTwin analysis at 35% of similarity

| Twin id[a,b] | Host(s) (n) | S. aureus subcluster(s)[c] | Gene family id | Function |
|---|---|---|---|---|
| 129 | S. aureus (29), S. pseudintermedius (12), S. haemolyticus (1), S. capitis (1), S. hominis (2) | B2 (29/29) | 255 | Tape measure chaperone |
| | | | 237 | Portal protein |
| | | | 246 | Head-to-tail connector |
| | | | 240 | Head morphogenesis protein |
| | | | 254 | Tail assembly |
| 141 | S. aureus (35) | B5 (35/35) | 92 | Terminase small subunit |
| | | | 66 | Transcriptional regulator |
| | | | 108 | Major capsid protein |
| | | | 166 | Tail tube protein |
| 338 | S. aureus (38), S. epidermidis (10), S. haemolyticus (1) | B3 (36/36)/B4 (2/2) | 467 | Tail assembly chaperone |
| | | | 463 | Head tail connector |
| | | | 465 | Tail completion protein |
| | | | 464 | Tail protein |
| | | | 431 | Head morphogenesis |
| | | | 466 | Major tail protein |
| 160 | S. aureus (18) | B6 (18/18) | 306 | Terminase small subunit |
| | | | 307 | Portal protein |
| | | | 94 | Terminase large subunit |
| 175 | S. aureus (14), S. sciuri (2) | B7 (14/14) | 620 | Tail assembly chaperone |
| | | | 631 | MazG_like_nucleotide_ pyrophosphohydrolase |
| | | | 98 | Portal protein |

[a]The twins with the highest number of gene family members are listed (core twins).
[b]id, identity.
[c]The number of phages of *S. aureus* subclusters for each twin is indicated.

were subjected to all-against-all BLASTP. Reciprocal hits with an E value of $<10^{-5}$ and mutual coverage of $>80\%$ were kept. The gene family genome bipartite graphs were constructed by using the similarity threshold of 35% and 95% in order to compare distant and more recent gene family transmission, respectively.

At 35% of similarity 1,484 gene families were generated and 346 twins (as mentioned above, twin is defined as group of homologue genes shared by a set of genomes) were identified; 22 out of 346 twins were nontrivial (composed by more than one gene family), whereas at 95% of similarity, 533 twins, with 480 trivial twins and 53 nontrivial twins, were identified.

The function of the core gene family was analyzed using the twins identified by MultiTwin with 35% similarity. The twins with the highest number of gene family members were included in this analysis (Table 2). Twin core 129 is composed of 5 gene families encoding proteins contained in phages from *S. pseudintermedius*, *S. haemolyticus*, *S. capitis*, *S. hominis*, and *S. aureus* subcluster B2 (29/29). Twin core 338, constituted of 6 gene families encoding proteins with different functions, is shared by *S. aureus* phages of subclusters B3 (36/36) and B4 (2/2), *S. epidermidis* phages (10/15), and an *S. haemolyticus* phage (1/3). Twin core 141 was present in 35 genomes of phages belonging only to *S. aureus* subcluster B5 (35/35), with no other staphylococci sharing it. As described above, this twin core contains proteins from four different families. The core twin 160, present in *S. aureus* phages grouping in subcluster B6, is formed by 3 gene families, while core twin 175 was shared by *S. aureus* phages of subcluster B7 (14/14) and 2 *S. sciuri* phages (2/2).

In order to analyze proteins with the highest conservation, the similarity threshold was set to 95%; in this way, 96.4% of the twins belong to unique hosts, and the remaining twins are shared by two or more hosts. Several clusters of phages have core twins composed mainly of genes that code for structural proteins (Table 3). Of note, all *S. aureus* phage subclusters have a core twin composed mainly of genes encoding packaging and structural proteins; an exception is phages falling into subcluster B3 in which twin cores 365 and 463 were present with nearly equal distribution (19/36 and 17/36, respectively). Our phages, namely, vB_SauS_287, vB_SauS_760, vB_SauS_I73, vB_SauS_Mh15, and vB_SauS_Mh1, are located in the last group (twin 463). Interestingly, the twin core 80 is composed of five gene families and is present only in phages grouped in subcluster B7; however, phage

**TABLE 3** MultiTwin analysis at 95% of similarity

| Twin id[a] | Host | Subcluster[b] | Gene family id | Function |
|---|---|---|---|---|
| 375 | *S. aureus* | B6 (18/18) | 543 | Clp_protease_like_protein |
| | | | 1705 | rinA |
| | | | 1712 | Portal_protein |
| | | | 516 | Terminase_large_subunit |
| | | | 1711 | Terminase small subunit |
| 192 | *S. aureus* | B2 (28/29) | 1390 | Tape measure chaperone |
| | | | 1152 | Minor capsid protein |
| | | | 1130 | Portal protein |
| 365 | *S. aureus* | B3 (19/36) | 2191 | Tail assembly protein |
| | | | 1513 | Phage tail protein |
| | | | 2172 | Putative major tail protein |
| | | | 2167 | Head tail protein |
| | | | 2186 | Major tail protein |
| | | | 1524 | Minor structural protein |
| | | | 2181 | Tail completion protein |
| 50 | *S. aureus* | B6 (17/18) | 1731 | MTP |
| | | | 1728 | HP |
| | | | 1713 | Major capsid protein |
| 80 | *S. aureus* | B7 (13/14) | 2463 | Nucleoside triphosphate pyrophosphohydrolase family protein |
| | | | 538 | Portal protein |
| | | | 510 | HNH endonuclease |
| | | | 2428 | Terminase small |
| | | | 2450 | HK97 protein family |
| 156 | *S. aureus* | B5 (35/35) | 699 | RBP |
| | | | 542 | Clp_protease_like_protein |
| | | | 544 | Major capsid protein |
| | | | 513 | Terminase small |
| | | | 611 | Major tail protein |
| | | | 691 | Tail endopeptidase |
| | | | 606 | Major tail protein |
| 463 | *S. aureus* | B3 (17/36) | 2194 | Tail assembly chaperone |
| | | | 2185 | DUF3168 protein |
| | | | 1523 | Tail endopeptidase |

[a]The twins with the highest numbers of gene family members are listed (core twins).
[b]The numbers of phages of *S. aureus* subclusters for each twin are indicated.

vB_SauS_308, belonging into this subcluster, is the only one excluded from this twin and, therefore, shows divergence from the other members.

In order to have a comprehensive, easy way to display the MultiTwin analysis, the bipartite networks at 35% and 95% protein similarity were plotted using Cytoscape, (Fig. 3A and B, respectively); visualization is enhanced by assigning color codes to staphylococcal species or subcluster classification. In the bipartite network (35%), the nodes corresponding to phage genomes were distributed in three main modules. The module 1 is composed of phages belonging to subclusters B2, B3, and B4; all the phages of *Staphylococcus pseudointermedius* and *S. hominis* and some phages of *S. epidermidis*, *S. haemolyticus*, and *S. capitis* are in a close relationship with module 1 (Fig. 3A and B). This finding is in agreement with the gene families shared by these phages (twin identity [id] 129 and 338) (Table 2). The module 2 is constituted by all the phages of subclusters B6 and B7 and related to phages of *S. sciuri*, as seen by twin 175, which is present in all of them. The module 3 is formed by phages of subcluster B5 and one phage of *S. epidermidis* that are closely related in spite of not sharing core twins. Finally, phages from *S. warneri* and *Staphylococcus saprophyticus* and from the only *S. aureus* member of the subcluster B17 show few shared genes and are located more distantly from module 3 (Fig. 3A and B).

When the analysis was performed at 95% identity, the network showed a much more segregated pattern, with well-defined cluster distribution of the genome nodes by host, except for the *S. epidermidis* phages that formed two distant clusters (Fig. 3). Neither the two phages of *S. sciuri* nor the two phages of *S. epidermidis* shared gene families with any other phage. The relationship of the subclusters maintained the same pattern as that at 35% identity, with phages of

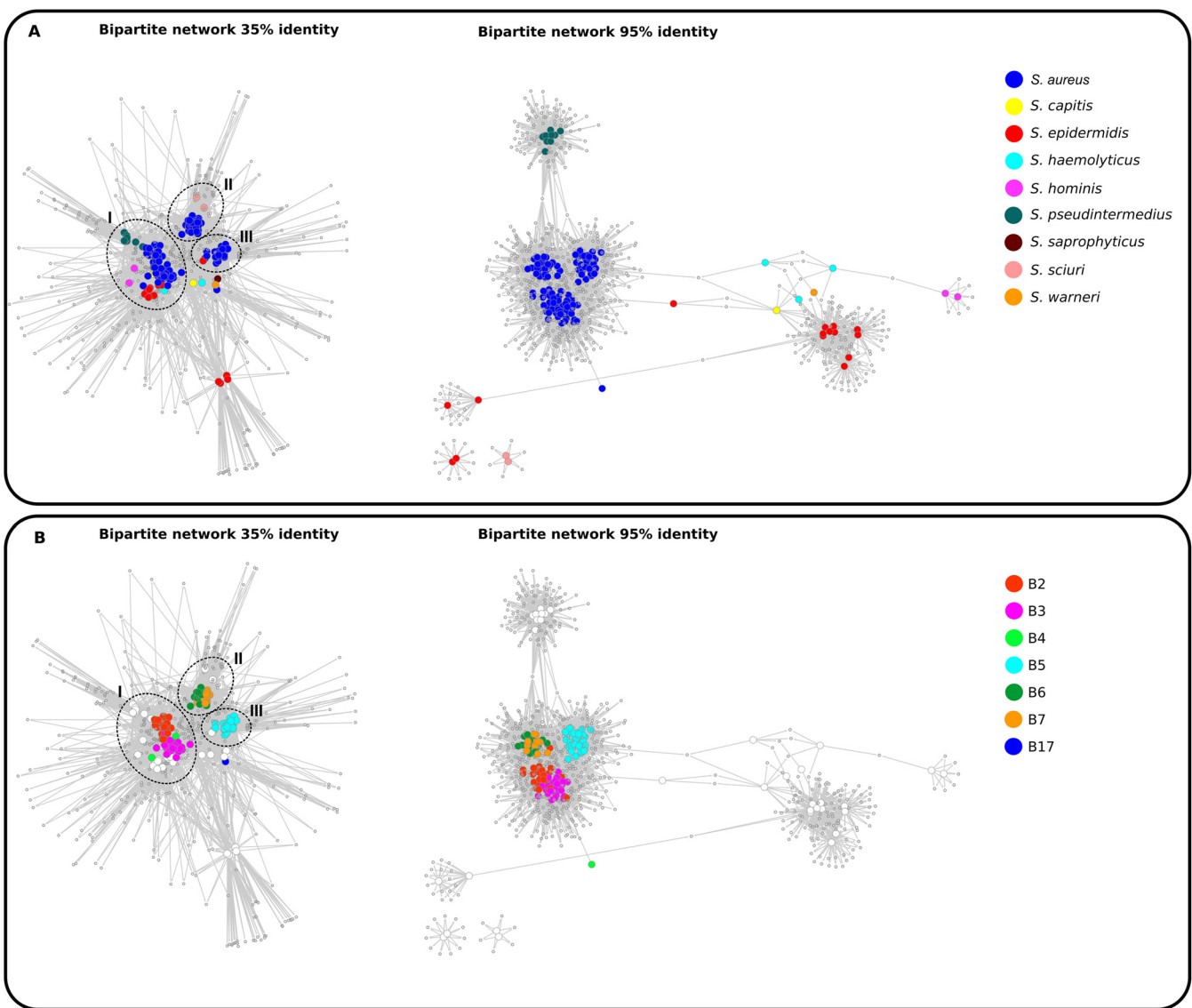

**FIG 3** Analysis of gene sharing using a bipartite network display. The network was calculated by the program MultiTwin (63), and a graph was generated using Cytoscape (64). The graph is composed by two types of nodes, namely, the genomes (colored) and the gene families (gray). (A) Displays the bipartite network colored by host, left at 35% of identity and right 95% of identity. (B) Shows the bipartite network colored by the *S. aureus* subcluster at 35% and 95% of identity (left and right, respectively). Genome distribution was grouped into modules (black circles).

subclusters B2 and B3 grouping together and phages of subclusters B6 and B7 grouping together separately. Phages of subcluster B5 were separated from the other subclusters, while phages from subcluster B4 were very distant from any other subcluster group showing its divergence from other phages. Phages from subcluster B17 did not hold to this level of identity stringency and thus are missing in this representation.

**Integration site integrase excision.** The analysis of the genomes of the phages reported here confirmed their temperate nature through the presence of genes encoding integrases and recombinases. Six of the 14 integrases belonged to the serine integrase (S-Int) family. The domain organization of phage S-Int has been described by Smith and Thorpe (14). These large proteins contain 4 discernible domains, as follows: first, a conserved N-terminal domain which contains the catalytic site; second, another conserved domain of 220 residues; third, a variable region of 125 amino acids; and fourth, a 30-amino-acid-long stretch enriched in branched amino acids or methionine that finishes in the C-terminal domain. Forty S-ints from phages in our database isolated from different staphyloccal species (28 *S. aureus*, 9 *S. epidermidis*, 2 *S. hominis*, and 1 *S. capitis* S-int) grouped in twin node 343. The amino acid alignment of

those S-ints showed a high homology, with a large set of 14 of them (including our 6 phages) almost 100% identical and phylogenetically related (see Fig. S1A and C in the supplemental material). The remaining 8 integrases belonged to the tyrosine integrase (Y-Int) family (Fig. S1B).

Sequencing analysis of the genes encoding the integrases has been used as a criterion for grouping *S. aureus* temperate phages, of which seven major groups, designated Sa1int to Sa7int, were proposed (15). In addition to those major groups, five singleton groups were designated Sa8int to Sa12int. Interestingly most of our integrases (6/14) fell into Sa7int-containing serine recombinases, while the remaining ones were distributed evenly into groups Sa1int to Sa6int (Fig. 4). This result is in agreement with the previous report by Goerke et al. (15) indicating that Sa7int was comprised only of serine recombinases. A remarkable exception on the integrase grouping was that of the integrase of phage vB SauS_Mh1 which fell into Sa9int, which is the second integrase described for this group. The analysis of the phage database we built showed that there are four additional members (phages Sebago, Henu2, StauST398-5, and StauST398-3) for the Sa9int group, and all these phages along with phage 96 belonged to subcluster B3.

**DNA packaging mechanism.** The DNA packaging and head module encodes the proteins required for efficient loading of the assembled head particles with genome copies. This system is composed of a portal protein forming a channel which allows for DNA entry and a terminase constituted from the TerS and TerL subunits, which are enzymes that are required for concatemeric DNA cleavage into genome size length units and for subsequent translocation of DNA into phage heads (16, 17). Esterman et al. (18) have predicted the packaging strategy by comparing the phylogenetic relationships between TerL proteins and the experimental evidence on packaging mechanisms available. We incorporated the TerL sequences of our phages to this analysis (Fig. 5A) and showed that the predicted mechanisms of 11/14 phages were that of a headful mechanism (P22) of DNA, characterized by the terminase recognition of *pac* sites. On the other hand, phages vB_SauS_320, vB_SauS_690, and vB_SauS_308 contain TerL proteins related to the HK97 TerL family, which is characteristic of phage genomes with cohesive ends (3'COS HK97, *cos*). In the case of vB_SauS_308, a prohead serine protease (gp50) is present in the DNA packing module; such a specific protease is present usually in *cos*-type phages (17). A scheme of the genetic organization of the DNA packing module is shown in Fig. 5B. A phylogenetic analysis showed that TerL proteins of phages using a *pac* mechanism were closely related, falling in two groups in accordance with their subcluster (B2 or B3). Phages using the *cos* mechanism were distantly located (Fig. 5A), whereas TerL of vB_SauS_320 and vB_SauS_690 (B5) were highly related and vB_SauS_308 (B7) TerL was located in a different branch. Thus, on the basis of our results, the packaging mechanisms are conserved between different clusters of phages.

**Lysis module.** Bacteriophages rely on the joint expression of the genes carried in the lysis module, namely, holin and endolysin, to successfully release the viral particles made during the replication cycle. Holins are pore-forming proteins that insert into the cytoplasmic membrane allowing the translocation of endolysins endowed with peptidoglycan hydrolase activity. Endolysins are characterized by the presence of a cell wall domain (CBD) and one or more enzymatically active domain (EAD) which act on peptide or glycosidic bonds of the peptidoglycan (19). In spite of the conservation of the function of these proteins, sequence divergence has been used as a way to classify staphylococcal phages. Thus, we analyzed both holin and endolysin genes and gene products in the phages under study. Holins displayed a gene length ranging from 255 to 438 nucleotides (nt) (438, 303, 276, and 255 nt), falling into the reported gene polymorphisms (486, 438, 435, 423, 303, 276, 273, 216, and 255 nt) of this gene (15).

Endolysins on the other hand were very homogeneous in length due to the presence of two EADs along with the CBD; in all the phages under analysis, both EADs consisted of a cysteine histidine-dependent amidohydrolase/peptidase (CHAP) domain along with an amidase domain (AMI) of either 2 or 3 type with the exception of the endolysin of phage vB_SauS_308 which contains only a CHAP EAD domain. The analysis of CBD domains revealed the presence of a majority (11/14) of SH3_5 sequences (20), with the remaining phages containing either V_CBD (*n* = 2) or I_CBD (*n* = 1) (Fig. 6). The grouping of our endolysins based on their gene

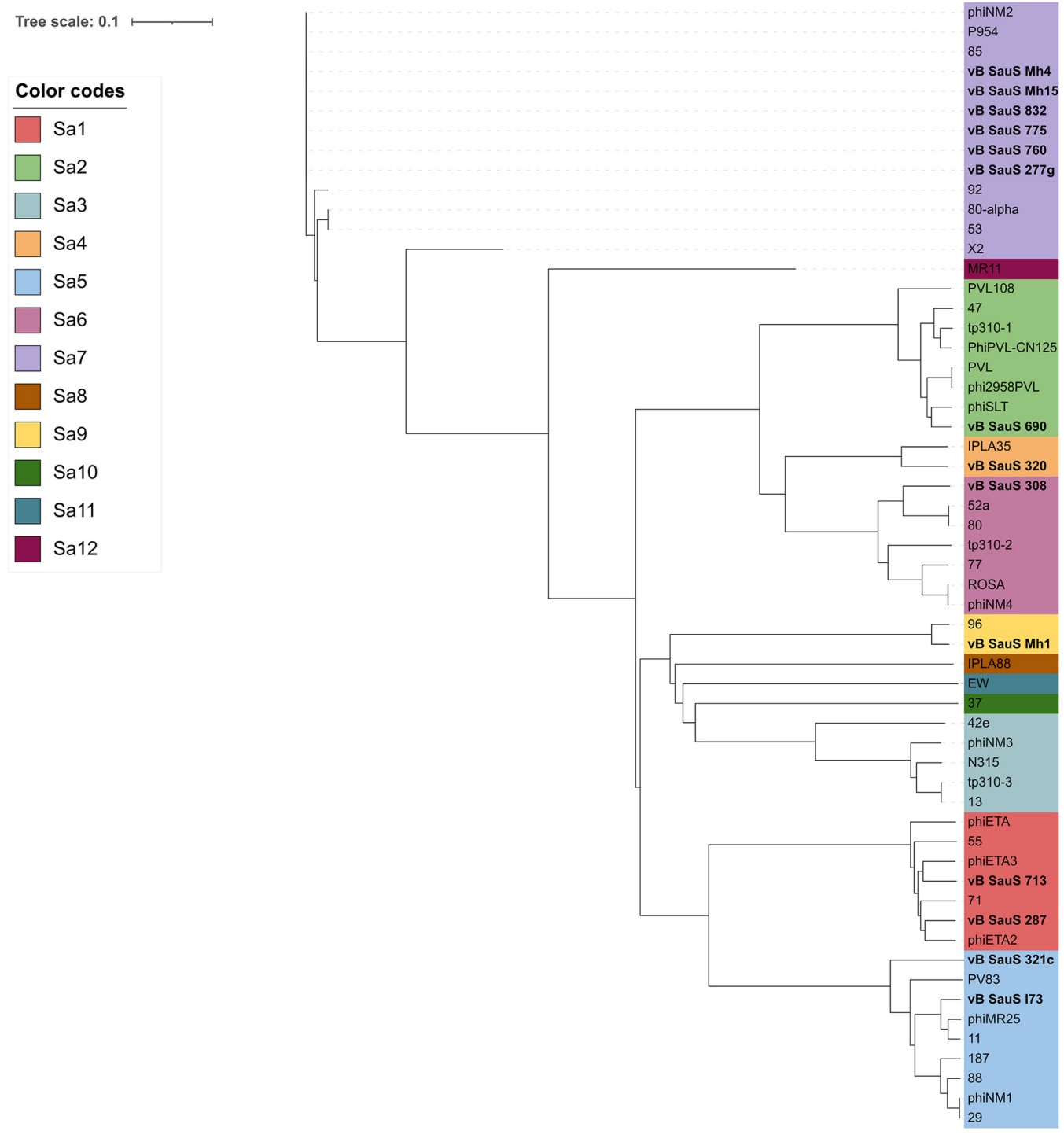

**FIG 4** Phylogenetic analysis of the *S. aureus* phage integrases. The classification of integrases in different groups (Sa1 to Sa12) was made on the basis of the nucleotide sequence comparison using ClustalW (65). The evolutionary history was inferred by means of the NJ algorithm, and the phylogenetic tree graph was made with iTOL (67). The integrases of phages reported in this work are shown in bold.

length, domain composition, and amino acid homology, based on work published previously by Chang and Ryu (21), showed that six phages contained group III endolysins (Fig. 6B and C) and holins of 438 bp, while five phages contained group IV endolysins (Fig. 6D) and holins of 303 bp. Two phages contained type V endolysins (Fig. 6E) with holins of 276 bp while only one phage (vB_SauS_308) contained a type I endolysin (Fig. 6A) and a 255-bp-long holin. The sequences of I_CBD and V_CBD are still unknown; however, a domain analysis indicates

**A)**

**B)**

**FIG 5** Prediction of the packaging strategy. (A) The phylogenetic relationships of the TerL proteins were obtained through protein sequence alignment using the MAFFT program (68) with the iterative method G-INS-i and FastTree program (72). (B) Scheme of the genetic organization of the DNA packing module of the 14 phages characterized during this work. HP, hypothetical protein; TerS, terminase small subunit; TerL, terminase large subunit; MCP, major capsid protein; HNH, homing endonuclease.

insignificant matching (E value of ≈0) with SH3 domains for V_CBD. Of note, the endolysin of phage vB_SauS_Mh15 contains two adjacent genes separated by a group I intron, a feature also displayed by the staphylococcal endolysins of phages X2, G1, and 85 as mentioned by Oliveira et al. (22) (Fig. 6C). Whether both adjacent genes are part of a single-endolysin-encoding gene would require experimental demonstration.

**Host interaction module.** The identification of the determinants of the phage-host interaction and the elucidation of the underlying mechanism(s) are fundamental to understand the role of phages as vehicles of staphylococci dissemination, evolution,

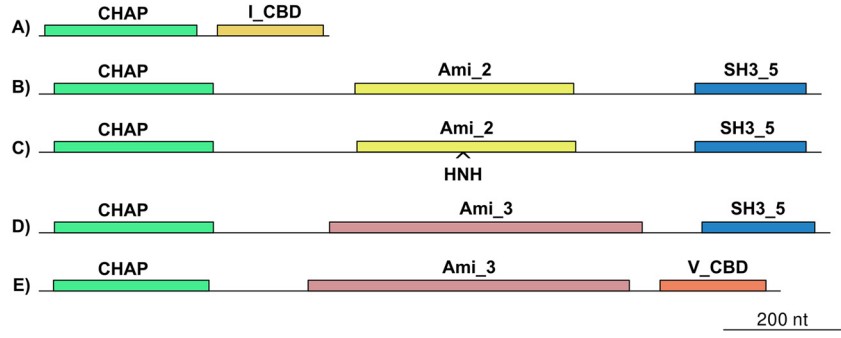

**FIG 6** Endolysin domain organization. CHAP, cystine histidine amidopeptide hydrolase; Ami_2 and Ami_3, Amidase_2 and Amidase_3, respectively; I_CBD, V_CBD and SH3_5, cell wall binding domains.

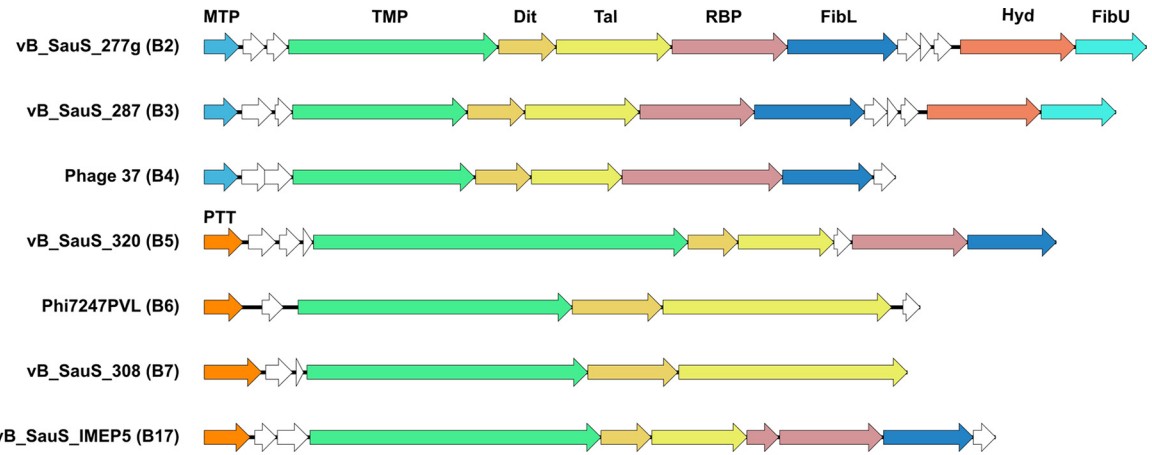

**FIG 7** Host recognition module organization. The organization of proteins hypothetically involved in the host recognition of phages of different subclusters is shown. For simplicity, only one representative phage of each subcluster was included. The figure was made using the EasyFig 2.0 program (72). MTP, major tail protein; TMP, tape measure protein; Dit, distal protein; Tal, tail associate lysin; RBP, receptor binding protein; FibL, tail fiber protein; Hyd, cell wall hydrolase; FibU, collagen-like fiber protein; PTT, phage tail tube protein.

and pathogenicity. Thus, we undertook the study of the phage module related to attachment to the host. The bioinformatic analysis was modeled on what is known for the well-studied staphylococcal phage $\phi 80\alpha$ (23). The proteins involved were major tail protein (MTP), tape measure protein (TMP), distal tail protein (Dit), tail-associated lysin (Tal), receptor binding protein (RBP), tail fiber protein (FibL), cell wall hydrolase (Hyd) and collagen-like fiber protein (FibU); the analysis of their interactions allowed for a detailed model of the structure (23). A scheme of the gene organization is displayed in Fig. 7. A genomic analysis of our phages identified the genes corresponding to tail and baseplate structures based on genome position, sequence homology, and function prediction using HHPRED and BLASTP (data not shown). As shown in Fig. 2, phage $\phi 80\alpha$ displays homology to other phages grouped in cluster B2; members of the clusters B2 and B3 maintain a similar organization of the components of host recognition, while phages belonging into clusters B5, B6, B7, and B17 differ in both gene identity and organization (Fig. 7).

In first place, we analyzed TMP, the largest protein in phage genomes, finding a wide distribution in protein sizes in the phages of our local library (675 to 2,757 amino acids); of note, phages falling into the B5 cluster have the longest TMP proteins, as shown in our phages vB_SauS_320 and vB_SauS_690 (2,066 residues and 2,074 residues, respectively) (Fig. 7 and 8).

An analysis of the phylogenetic relatedness among TMPs showed that they grouped in two major branches, as follows: one includes phages from clusters B6, B7, B17, and B5 along with phages infecting *S. epidermidis* and *S. sciuri*, while the second one includes members of clusters B3, B4, and B2 as well as phages infecting *S. epidermidis*, *S. haemolyticus*, *S. capitis*, *S. hominis*, and *S. pseudintermedius* (Fig. 8). Of note, relatedness among members of this second group is higher as clustering is more homogenous.

The most relevant protein upstream of TMPs is the major tail protein (MTP). Proteins located upstream of TMP, with homology to the MTP of phage $\phi 80\alpha$, are present in phages belonging to subclusters B2, B3, and B4; however, phages forming part of subclusters B5, B6, B7, and B17 lack this protein, encoding a phage tail tube protein (PTT; PF04630) instead.

Genes encoding the distal tail protein (Dit) were also detected in our phages, which was an expected result as Dit is conserved in phage tail tips (24); we found that gp61 of phage vB_SauS_308 encodes the largest Dit protein described in our set of phages (see Table S3 in the supplemental material). vB_SauS_308 gp61 displays homology with Dit proteins identified in phages belonging to subclusters B7 and B6 (data not shown); as seen in Fig. 7, the protein length in also comparable in these three subclusters.

A detailed analysis of gp61 detected three domains, namely, N (NTD), central (CD), and C (CTD). The gp61 vB_SauS_308 NTD revealed high homology to Dit (gp58) of $\phi 80\alpha$, while

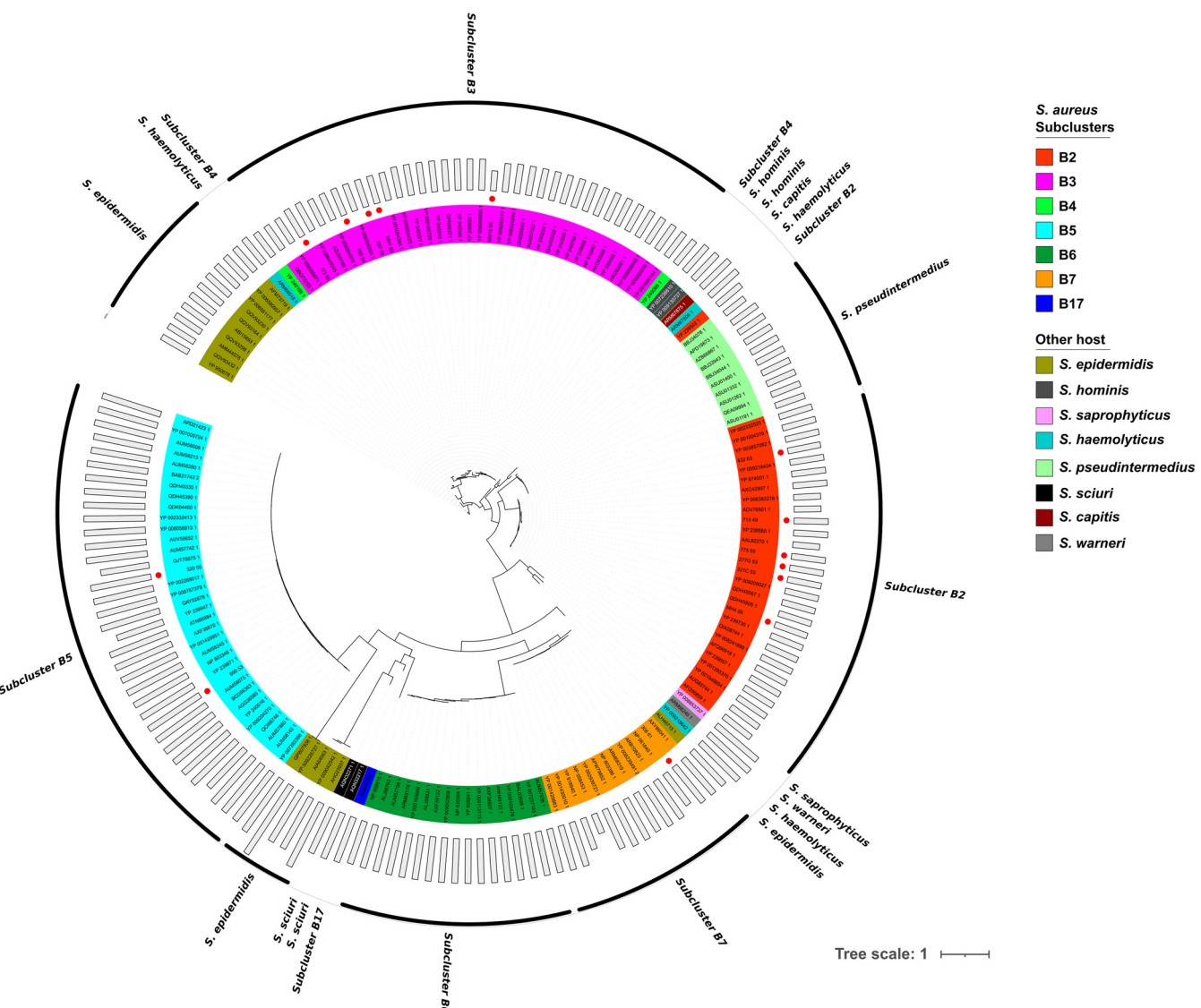

**FIG 8** Phylogenetic analysis of tape measure proteins (TMPs). The amino acid sequences of 169 TMPs were aligned using MAFFT (68), and the phylogenetic tree was estimated by applying the maximum likelihood algorithm with the PhyML program (70). The representation of the tree was made with iTOL. The outermost bars represent the length (amino acids) of each TMP protein. *S. aureus* subclusters or other non-*S. aureus Staphylococcus* hosts are identified with different colors as shown on the right side of the figure. TMPs of phages described in this work are marked by red dots.

CD displayed structural homology to 5LY8_A, which was reported as being involved in carbohydrate binding in *Lactobacillus casei* phage J1 (25). Finally, the CTD showed homology to 2X8K_B, a domain described in the Dit protein of phage SPP1 from *Bacillus subtilis* (26). Phage vB_SauS_308 is also outstanding because of its Tal protein which is much bigger than Tal proteins of other phages; this feature is also shared by phages in subclusters B6 and B7 as shown in Fig. 7. The Tal protein from phage vB_SauS_308 has a CTD with homology to intramolecular chaperones (3GW6_C) which were described in an endosialidase of *Escherichia coli* phage K1F and in a structural (neck appendage) protein of *Bacillus subtilis* phage GA-1 (27); it also has an NTD homologous to other Tal proteins (PDB 6V8I_BE).

Proteins with evident homology to hydrolases (esterases, lipases, or peptidoglycan hydrolases, such as gp67 in phage $\phi80\alpha$) were not detected in phage vB_SauS_308, which is a commonality with phages belonging to clusters B6 and B7. These phages have also a distinctiveness expressed as the lack of RBP, FibL, Hyd, and FibU, suggesting uniqueness in the mechanism for host recognition.

All the members of the cluster B5, including phage vB_SauS_320, display a similar gene organization, containing TMP-, Dit-, Tal-, RBP-, and FibL-encoding genes but lacking Hyd- and

**TABLE 4** Analysis of the correlation of endolysin type, holin length, and integrase

| Type_endoysin | Holin length (nt[a]) | Integrase group | Phage[b] |
|---|---|---|---|
| Group_I | 255 | Sa3 | 13, 42e, N315, phiNM3, tp310-3 |
| | | Sa5 | 187, PV83 |
| | | Sa6 | **vB_SauS_308** |
| Group_III | 438 | Sa1 | 55 |
| | | Sa5 | 11, 29, 88, phiMR25, **vB_SauS_321c, vB_SauS_I73** |
| | | Sa6 | 52a, 80 |
| | | Sa7 | 92, phiNM2, **vB_SauS_760, vB_SauS_Mh4** |
| | | Sa9 | **vB_SauS_Mh1** |
| | | Sa12 | MR11 |
| | | Sa7 | **vB_SauS_Mh15** |
| | | Sa7 | X2 |
| | | Sa7 | 85 |
| | | Sa8 | IPLA88 |
| | 423 | Sa10 | 37 |
| | 435 | Sa11 | EW |
| Group_IV | 303 | Sa2 | 47, phi2958PVL, PhiPVL-CN125, phiSLT, PVL, PVL108, tp310-1, **vB_SauS_690** |
| | | Sa4 | IPLA35, **vB_SauS_320** |
| | | Sa5 | phiNM1 |
| | | Sa6 | 77, phiNM4, ROSA, TP310-2 |
| | | Sa7 | 53, 80-alfa, **vB_SauS_277g, vB_SauS_775, vB_SauS_832** |
| | | Sa9 | 96 |
| Group_V | 276 | Sa1 | 71, phiETA, phiETA2, phiETA3, **vB_SauS_287, vB_SauS_713** |

[a]nt, nucleotides.
[b]Phages isolated throughout this work are shown in bold.

FibU-encoding genes; however, we cannot discard the possibility that proteins containing other domains with low homology/different structure may replace those roles (Fig. 7).

**Integrases, endolysins, and holins as markers for phylogenetic relatedness.** One of the first described approaches for the analysis of genetic relatedness of staphylococcal temperate phages was based on the analysis of the clustering of the integrases which defined 12 groups (15). As shown above, when we included the groups determined by holin polymorphisms and endolysin gene length and domain composition, we found a correlation between integrases, holin polymorphisms, and endolysin group in groups Sa1int to Sa3int (Table 4). It is interesting that endolysins show a correspondence with holin length, a feature that will require functional studies to determine whether there is a functional reason for that finding.

**Biological characterization of selected phages.** In order to determine biological features of the isolated phages, we determined their host range; thus, phages vB_SauS_I73, vB_SauS_Mh15, vB_SauS_Mh4, vB_SauS_832, vB_SauS_308, and vB_SauS_320, which were taken as representatives of subclusters, B2, B3, B6, and B7, were tested for their ability to propagate on several different coagulase-negative staphylococci (CNS) isolated in a clinical bacteriology unit. Our results showed that none of the phages were able to propagate or the alter growth of *S. epidermidis* or *S. hominis* (four strains each); *S. haemolyticus*, *S. capitis*, or *Staphylococcus caprae* (two strains each); or *S. sciuri* or *Staphylococcus lugdunensis* (one strain each) with the exception of two strains of *S. hominis* that allowed for the propagation of all the tested phages. Interestingly, both *S. hominis* strains developed faint haloes at 18/24 h with more visible haloes and isolated plaques at 48 h (Fig S2). A further analysis demonstrated that the observed delay was due to the poor adsorption of the phages, with negligible binding (<2%) at time points lower than 1 h; in contrast, adsorption rates for *S. aureus* RN4220 were of 90% to 95% in the first 10 min (data not shown).

Serial 10-fold dilutions of each of the 6 selected phages formed plaques when spotted onto plates of *S. aureus* RN4220 that was lysogenic for each phage of the set, thus indicating that no obvious interference with phage entry or cross-resistance through shared immunity took place.

None of the phages were able to transduce plasmids pCN51 or pCN57 from the donor strain; however, $\phi$ 11, a well-known transducing phage used as a control, yielded

transductants at frequencies of $5 \times 10^3$ CFU/mL lysate (pCN51) and $3 \times 10^3$ CFU/mL lysate (pCN57) (average of three biological assays), which is comparable to those reported in the literature (28).

## DISCUSSION

Bacteriophages shape the genomic bacterial landscape, and such a role is of outmost importance in the case of bacterial pathogens, such as *S. aureus*, where temperate phages can be hijacked by mobile genetic elements encoding pathogenicity determinants (SAPIs) and, thus, be the drivers of the horizontal dissemination of those genes (10). Importantly, temperate *Siphoviridae* phages account for the majority of the known phages in staphylococci (29). The increase of sequenced *Siphoviridae* phage genomes over the last decade gave a large amount of information for bioinformatic analysis, which in turn prompted for the development of strategies to classify those phages in order to drive conclusions about evolutionary relatedness and distribution in clinical isolates. The first approach for a classification of *staphylococcus Siphoviridae* phages was based on gene polymorphisms of integrases and holins as reported by Goerke et al., with 12 groups (Sa int1 to Sa int12), of which 4 were the most heavily populated (15). Simultaneously Deghorain et al. grouped phages, which were isolated mostly from *S. aureus*, in 3 groups based on genome size, gene content, and protein content similarities (17). More recently, in a very comprehensive study, Oliveira et al. analyzed 205 sequences of staphylococcal phages of different origins that are available in public databases and applied Phamerator, which was developed for the bioinformatic analysis of mycobacterial phages (30), to link protein family relatedness (11). Thus, the bioinformatic analysis allowed for the prediction of 20,579 encoded proteins, sorted into 2,139 "phamilies" (designated "phams") of related sequences. Gene content analysis grouped phages into 4 clusters and 27 subclusters (A1 to A2, B1 to B17, C1 to C6, and D1 to D2) of which cluster B harbored 132 *Siphoviridae* phages, and most of them predicted to be temperate; importantly, several phages also contained genes encoding putative toxins and bacterial virulence factors. The most heavily populated subcluster is B2 ($n = 19$), followed by subclusters B3 ($n = 26$), B5 ($n = 26$), B6 ($n = 18$), and B7 ($n = 12$); host preference was evidenced by the fact that members of the subclusters B1 to B2 and B3 to B7 were isolated only from *S. aureus* or *S. pseudintermedius*, while members of the subcluster B4 could use *S. aureus*, *S. haemolyticus*, and *S. epidermidis* as hosts (11). Other coagulase-negative staphylococci (*S. sciuri*, *S. warneri*, *S. saprophyticus*, *S. haemolyticus*, and *S. hominis*) harbored phages grouping into the much less populated subclusters B8 to B17.

Through our work, we are adding one of the largest sets of sequenced staphylococcal *Siphoviridae* phages ($n = 14$) isolated from human and animal sources reported from a single laboratory. Our approach to perform the bioinformatic analysis was based on MultiTwin, an open access software suite that uses multipartite graphs to analyze evolution at multiple levels of organization (31). It has been used to study evolutionary connections between archaea viruses and mobile genetic elements, prompting us to apply it to study the sharing of gene families between phage genomes (32). Thus, to our knowledge, this is the first time MultiTwin is used to analyze *Staphylococcus* phage evolutionary relationships. Our results show that more than 70% of the gene content could be assigned to known proteins, of which almost 50% are functionally involved in DNA packaging, virion structure, cell lysis, lysogeny, or DNA replication. Subcluster assignation showed that 5 phages fell into subcluster B2, 6 into subcluster B3, and 2 into subcluster B5 and only 1 (vB_SauS_308) grouped with phages of subcluster B7. Thus, our analysis matches previous analyses pointing out that subclusters B2 and B3 are two of the most heavily populated and evolutionarily related subclusters (11).

Our bioinformatic analysis showed that all of them were temperate bacteriophages with the identification of integrases of the serine or tyrosine type, excisionases, and LexA repressors. The fact that the sequence phages reported here have been obtained by prophage induction with mytomicin C provides experimental proof of their temperate nature. Overall, integrase polymorphisms in our phages did not correlate to subclustering in an obvious manner; i.e., integrases of vB_SauS_321c (subcluster B2) and vB_SauS_I73 (subcluster B3)

correspond to Sa5 as classified by Goerke et al. (15); these results have also been pointed out by Oliveira et al. recently by using *pham* analysis (11).

Importantly, opposite to what has been described by these authors, we found very few toxin- and virulence-related genes in our set of phages. According to Oliveira et al. (11) virulence factors encoding genes are absent in most of the staphylococcal phage subclusters analyzed (B1, B4, B9, B10, B12, B13, B14, B16 and B17); are present in low numbers (roughly 30% in phages of subclusters B2 and B3); or are present in high numbers, such as in subclusters B5 (96% of the phages), B6, and B7 (100% of the phages). Genes encoding three known virulence factors, namely, MazF, VirE, and the metallo/endopeptidases family ImmA/IrrE, were detected in our set of phages. Bioinformatic analysis of our phage database showed that MazF was encoded in 24 *S. aureus* phages belonging into subclusters B2, B3, B6, and B5, as well as in 1 *S. hominis* phage. VirE was present in 34 *S. aureus* phages (subcluster B5); finally, metallo-endopeptidases of the ImmA/IrrE family (PF06114.16) were present in phages of *S. aureus* subclusters B5, B3, B2, and B4 ($n = 47$); *S. epidermidis* ($n = 9$); *S. pseudintermedius* ($n = 6$); and *S. capitis*, *S. haemolyticus*, and *S. hominis* ($n = 1$ in each case). Interestingly, phage vB_SauS_308, falling into subcluster B7, differs from the rest of the members of this subcluster as it lacks any virulence gene. Although speculative, we hypothesize that the low frequency of virulence factor genes observed in our phages could reflect a source bias during our screening, as many of the strains used were isolated from hands of healthy carriers (15).

By combining the bioinformatic analysis of TerL in our phages and the strategies for DNA packaging that had been experimentally validated, we predicted which strategy our phages would use, concluding that phages belonging to subclusters B2 and B3 follow a P22-like mechanism (headfull); this finding confirms the link between these subclusters (Fig. 3A). In the same way, we predicted the *cos* mechanism used by members of the subclusters B5 and B7.

Endolysins present in staphylococcal phages are encoded mostly in single genes containing 1 or 2 EAD domains along with a CBD of either SH3_5b, I_CBD, or V_CBD domains (175/205), with fewer cases in which an intron splits the gene (20/205); rarely, 2 separate genes have been described as encoding endolysins (9/205) (11). The analysis of the endolysins present on our phages confirmed that trend, with 12/13 endolysins encoded by a single gene versus 1/13 in which 2 adjacent genes are present (vB_SauS_Mh15). The frequency of EADs correlates with previous descriptions, showing the presence of a CHAP domain accompanied in all cases by either ami-2 or ami-3 domains in equal proportion. When holin length polymorphisms were analyzed, we confirmed a correlation between integrase polymorphisms, holin length, and endolysin groups as has been reported (15). However, this relationship held true only for the major integrase groups and precluded a general conclusion.

A subset of six phages (vB_SauS_I73, 308, 320, 832, Mh4, and Mh15), representing the different subclusters, were used to determine their host range on a set of 16 CNS clinical strains. Surprisingly, all the phages were able to adsorb, although at very low rate, and propagate in two *S. hominis* strains. There are very few *S. hominis* phages described; however, our initial analysis of one of them, namely, phage StB27 (GenBank accesion no. NC_019914), revealed that its Tal protein (gp44) belongs to Twin node 263 which also includes vB_SauS_277g, 287, 321C, 713, 760, 775, 832, I73, Mh4, Mh1, and Mh15. However, the *S. hominis* protein is shorter than the *S. aureus* homologs and distantly related in sequence (C.A.S., unpublished observation). Likewise, FibL falls into Twin node 180 which has 216 members, among them, our phages vB_Sau_277g, 287, 320, 321C, 690, 713, 760, 775, 823, I73, Mh1, Mh15 and Mh4 (C.A.S., unpublished observation). Thus, the overall organization of the receptor module of *S. hominis* phages may overlap enough with that of *S. aureus* phages to allow limited cross-infection in spite of the differences in cell wall teichoic acid composition (33). In addition, the fact that *S. aureus* lineage sequence type 395 (ST395) contains poly-glycerol-phosphate (GroP) wall teichoic acid (WTA) glycosylated with *N*-acetyl-D-galactosamine (GalNAc) instead of the most commonly found poly-ribitol-phosphate (RboP) substituted with D-alanine and *N*-acetyl-D-glucosamine (34) suggests that changes of WTA and/or phage receptors may be more common than supposed, thus allowing for a back and forth journey of phages between CNS and *S. aureus* strains as postulated by Deghorain (35).

The understanding of the mechanisms underlying phage-host interactions is of outmost importance if phages are proposed as biocontrol materials (for a recent review, see Moller et al., [36]). The propagation of phages may be blocked through the modification of cell envelope receptors (37, 38), clustered regularly interspaced short palindromic repeat (CRISPR), or restriction-modification (R-M) systems (36). Importantly, a pending assignment is the study of the mechanisms of superinfection exclusion that may be pivotal for the survival of a lysogenic *S. aureus* strain when encountering staphylococcal phages. Studies on that direction have been undertaken in the system *Mycobacterium*-mycobacteriophage (39). A passive mechanism, such as envelope alterations, also implies oftentimes fitness challenges for *S. aureus* (38) and most likely niche alterations since the mobilization of SaPIs may be compromised (6). Moller et al. reported recently a species-wide genome sequencing that pinpointed a few (six) bacterial genes involved in host range determination (36); clearly, the limited set of genes that may be of importance to control phage susceptibility is many times less important than the gene evolution that would dictate the ability of a phage to infect resilient strains.

The overall analysis of our results reinforces the recent suggestion by Moller et al. (36) that the more phages we characterize, the faster we will define the ever-evolving strategy of the host-prey interaction. In this direction, recent work by Göller et al., reporting the isolation of 94 staphyloccoccal phages from the environment (of which 40 were sequenced and analyzed) on a multispecies *Staphylococcus* cocktail as the host, demonstrated that there is less species stringency for phage infection and propagation than thought previously (40). Although the analysis cannot directly be extrapolated to our set of strains and phages, it is quite enticing thinking that the role of cell WTA may not be a stringent determinant for establishing the phage host range. In this sense, the twin analysis we report here shows proteins involved in host recognition which are shared by phages infecting different species, which is a result that supports the conclusions made by Göller and Deghorain.

Interestingly, a recent report addressing the bioinformatic analysis of 211 prophages contained in the genomes of 58 *S. aureus* strains isolated from a very defined niche—patients suffering from chronic rhinosinusitis (CRS)—revealed the presence of *iec*, the human immune evasion cluster (41); this cluster was not present in our phages, stressing the vastness of the yet-uncharted territory of host and niche adaptation through temperate phage gene supply.

With no doubt, the field of staphylococcal phage genomics will advance at a faster speed from now on, based on the low-priced sequencing costs and bioinformatics methods available for developing and analyzing networks of relationships at a very high scale.

## MATERIALS AND METHODS

**Growth media, chemicals, strains, and growth conditions.** *Staphylococcus aureus* strains and coagulase-negative staphylococci (CNS) were grown routinely in tryptic soy broth (TSB) or tryptic soy agar (TSA), supplemented with 2 mM $CaCl_2$ and 10 mM $MgSO_4$ when used for bacteriophage detection or propagation as a liquid or solid medium. Cultures were incubated at 30°C and 60 rpm for chosen times while plates were incubated at 30°C for 24 h. Chemicals (analytical grade or better) and enzymes were purchased from Sigma (St. Louis, MO) or from local vendors. Lab stock strain *S. aureus* RN4220 containing plasmids pCN51 or pCN57 (the generous gift of R. P. Novick, New York University [NYU], USA) (5) were grown with erythromycin (10 $\mu$g/mL).

**Bacteriophage techniques.** Bacteriophages were obtained from *S. aureus* strains isolated from different human and animal sources as described in Table 1. We searched for bacteriophages that were either spontaneously released or induced by treatment with mitomycin C. In the first case, nasal samples taken with cotton swabs from human nares were pooled in groups of 10 swabs, and to each group, 4 mL of phage buffer (PhB; 2 mM $CaCl_2$, 10 mM $MgSO_4$, 50 mM Tris HCl [pH 7.6], and 150 mM NaCl) was added in centrifuge tubes. After an overnight incubation at room temperature, the cotton swabs were removed, the tubes were centrifuged (3,000 rpm, 10 min, and 4°C), and the supernatant was transferred to a clean tube. In the case of prophage induction, mitomycin C (0.5 $\mu$g/mL) was added to early-log phase (optical density at 600 nm [$OD_{600}$], 0.2) cultures of each *S. aureus* strain grown in TSB; after 6 h of incubation (30°C, 60 rpm), the treated cultures were centrifuged (3,000 $\times$ g, 10 min, and 4°C) and the supernatant transferred to a clean tube and neutralized to pH 7 with 0.1 M NaOH. Finally, each supernatant was filter sterilized through filters with a 0.45-$\mu$m pore size and kept at 4°C.

Phage detection was made using *S. aureus* RN 4220 as indicator strain and the double layer agar technique (42); briefly, 100 $\mu$L (spontaneous release) or 200 $\mu$L (mitomycin C induction) of each sample was added to 100 $\mu$L of an overnight culture of RN 4220 (grown at 30°C in TSB) and incubated at room

temperature for 20 min. Finally, molten top agar (4 mL, 0.4 [wt/vol] in TSB) at 55°C was added to each sample; upon mixing, each sample was poured on top of TSA plates. Upon hardening, plates were incubated at 30°C for 24 h. Individual lysis plaques (one from each plate excepted when large differences in size of morphology were observed) detected by visual observation were picked and isolated three times using standard procedures. Phage amplification and titration of lysates was performed as described previously (42).

**(i) Host range and phage adsorption assays.** Sixteen isolates of CNS species (4 *S. epidermidis*, 4 *S. hominis*, 2 *S. caprae*, 2 *S. haemolyticus*, 2 *S. capitis*, 1 *S. lugdunensis*, and 1 *S. sciuri* isolate, which were kindly provided by M. Almuzara and C. Vay, Bacteriology Service, Hospital de Clínicas, Universidad Nacional de Buenos Aires) were identified by phenotypic tests and matrix-assisted laser desorption ionization–time of flight mass spectrometry (MALDI-TOF MS). The host range displayed by the phages under testing was determined by mixing 4 mL molten top agar with 100 $\mu$L of overnight cultures of each CNS and pouring the mixture on top of TSA plates. After hardening, 10-fold dilutions of each phage on PhB were spotted on top of each plate, followed by incubation for 24 to 48 h at 37°C. Upon visual inspection, the presence of haloes and the number of plaques were recorded. When necessary, phage adsorption was measured as described by Uchiyama et al. (43). Briefly, 200 $\mu$L of each culture was diluted with 200 $\mu$L of TSB medium in microcentrifuge tubes, followed by addition of 100 $\mu$L of the phage being tested at a multiplicity of infection (MOI) of 0.01. After a gentle mixing step at room temperature, 100-$\mu$L aliquots were withdrawn at 1, 5, 10 and 15, and 30 and 60 min; set on ice; and centrifuged (10,000 rpm, 1 min). Finally, phage titer in the supernatants was determined as described above. The assays were made in biological triplicates.

**(ii) Immunity tests.** Lysogens of *S. aureus* RN4220 were obtained, and its lysogenic nature confirmed by UV or mitomycin induction for each phage of interest as described above. Immunity tests were performed by spotting serial dilutions of each phage onto plates containing each of the *S. aureus* RN4220 lysogens. Plates were observed visually after 24 h at 37°C.

**(iii) Transduction assays.** The transducing ability of the phages isolated throughout this work was analyzed following the protocol described by Olson (44). To that end, *S. aureus* RN4220 transformed with plasmids pCN51 (Ery$^r$) or pCN57 (Ery$^r$, *gfp*,) were used as the donor in transduction assays. Transducing phage $\phi$11 (45, 46) (kindly provided by L. Marrafini, Rockefeller University) was used as a positive control for these assays. Plasmid transductions were selected by antibiotic resistance and confirmed by the detection of fluorescence using an Olympus MVX10 binocular scope. The frequency of transduction was expressed as the number of Ery$^r$ or Ery$^r$ *gfp*$^+$ colonies/mL transducing lysate ($3 \times 10^{10}$ PFU/mL).

**DNA extraction and genome sequencing.** DNA was extracted from high-titer lysates ($10^9$ to $10^{11}$ PFU/mL) and treated with DNase I (1 $\mu$g/mL) and RNase (1 $\mu$g/mL) by using guanidine thiocyanate and the Wizard DNA clean-up system (Promega) as described previously (42). DNA concentration was determined spectrophotometrically by measuring $A_{260}$. Purified DNA was tested for integrity by agarose gel electrophoresis and kept at $-20$°C until use.

Phage genome sequencing was carried out at a local facility (Instituto de Agro-Biotecnología de Rosario [INDEAR], Argentina) using the Illumina HiSeq 1500 platform, and library preparation was performed as indicated by the manufacturer (Nextera XT; Illumina Inc., San Diego, CA).

**Bioinformatic analysis. (i) Annotation and genome organization.** Phage hypothetical open reading frames (ORFs) of all genomes were annotated using DNA Master (http://Phagesdb.Org/DNAMaster/), Glimmer (version 3.02) (47), and GeneMarkS (version 4.28) (48) and were curated manually. Putative functions were assigned using BLASTP, HHPred (49), Pfam (50), and tRNAscan-SE (51) in search of tRNAs. IslandViewer online software (52) was used to predict genomics islands. Phages were named using the nomenclature of viruses proposed by Lavigne et al. (53). Genome features were extracted or calculated with the help of packages Biostrings (54) and GenomicRanges (55) from Bioconductor in R. Genome maps were created using genoPlotR version 0.8.11 (56) running in R Software version 3.6.3 (57). Virulence factors were detected with VFanalyzer within the database of factors of virulence, which was accessed online through http://www.mgc.ac.cn/cgi-bin/VFs/v5/main.cgi?func=VFanalyzer (58).

**(ii) Bacteriophage classification.** The taxonomic classification of our phages was performed using vConTACT2 executed on the Cyverse platform (https://cyverse.org/). As an input, we used a multifasta file containing all proteins encoded by our phages and a gene-to-genome mapping file. Diamond was used to calculate protein-protein similarity with a E value threshold of $1e^{-5}$, while virus cluster (VC) was generated using the Cluster ONE method. For this analysis, the NCBI Bacterial and Archaeal Viral RefSeq V85 was employed (13, 59).

The subcluster classification of our phages as well as that of others that were not included in the analysis performed by Oliveira et al. (11) was carried out. First, we used Gegenees (60) to perform a fragmented alignment of 187 phage genomes; then, the outcome was employed to construct a phylogenetic network by using the Neighbor-Net method in the Splitstree 4.0 program (61).

**(iii) Bipartite network.** The interaction between gene families and phage genomes was performed by MultiTwin software (63). Gene sharing was evaluated through the identification of twin nodes. The term twins refers to a set of cluster of homologous gene nodes that have identical connectivity to genome nodes, which means gene families that are shared by the same sets of genomes (62). The construction of gene families was achieved by executing an all-against-all BLAST of the whole set of proteins encoded by the 173 phage genomes (available in public databases) included in this analysis, in addition to our 14 phages. The sequence similarity threshold was established at 35% and 95% with >80% of mutual coverage and an E value of $\leq 10^{-5}$. The bipartite network was plotted using Cytoscape version 3.8.0 (63).

**(iv) Analysis of the lysogeny module.** Nucleotide sequences of the integrases present in prophages contained in 44 *S. aureus* complete genomes were retrieved from the nucleotide database of NCBI (https://www.ncbi.nlm.nih.gov/nuccore) and analyzed together with the 14 sequences from our laboratory

using ClustalW for multiple sequence alignment (64). Then, a phylogenetic neighbor-joining (NJ) tree of similarity was constructed using the SeqinR package on R software and plotted using iTOL (version 6.4) (65, 66) to show the classification into different types. In addition, a multiple sequence analysis of the amino acid sequences of the integrases of our phages was achieved using MAFFT (67). A plot was made using Jalview version 2.11.1.7 (68). The evolutive relationship of serine integrases was inferred using the maximum likelihood algorithm in PhyML (69); the phylogenetic tree was plotted with iTOL (66).

**(v) Analysis of the lysis module.** The protein domain composition of our phage endolysins was analyzed using Pfam (50) with a search cutoff E value of <1 and grouped in different types according to Chang and Ryu (21). Holin sequence length (obtained from annotated phage genomes) polymorphism was calculated manually.

**Analysis of the DNA packaging module.** The mechanism of DNA packaging was predicted by comparing the amino acid sequence of the TerL proteins of our phages against those with experimental evidence for each packaging mechanism (18). The 87 TerL sequences were aligned with MAFFT using the iterative method G-INS-i (67), and then the informative regions of the alignment were estimated by BMGE version 1.2 using an entropy value of 0.8 (70). Afterward, a phylogenetic tree was built using the FastTree program (71). We assumed that if phages published elsewhere, having experimentally demonstrated DNA packaging mechanism(s), shared cluster grouping and TerL proteins with the phages reported here, it would allow us to safely propose a common packaging mechanism.

**(v) Analysis of the host interaction module.** The protein function of components of this module was evaluated by comparing protein structures generated through HHPred, PDB, and Pfam/A databases. The module organization was generated using EasyFig version 2.2.2 (72). Tail measure proteins (TMPs) were analyzed using a local database composed of 169-amino-acid sequences of TMPs that were aligned using MAFFT version 7 and G-INS-i iteration method using WSP and consistency scores, followed by an assessment of the phylogenetic informative regions by BMGE version 1.2 (70), with an entropy value of 0.8. The phylogenetic tree was made using PhyML under the maximum likelihood principle. Graphics were edited with iTOL.

**Data availability.** Genomic sequences of the phages reported here have been deposited in GenBank under the indicated accession numbers (vB_SauS_277g, OM439662; vB_SauS_287, OM439663; vB_SauS_308, OM439664; vB_SauS_320, OM439665; vB_SauS_321c, OM439666; vB_SauS_690, OM439667; vB_SauS_713, OM439668; vB_SauS_760, OM439669; vB_SauS_775, OM439670; vB_SauS_832, OM439671; vB_SauS_I73, OM439672; vB_SauS_Mh1 OM439673; vB_SauS_Mh15, OM439674; and vB_SauS_Mh4, OM439675).

## SUPPLEMENTAL MATERIAL

Supplemental material is available online only.

**SUPPLEMENTAL FILE 1**, PDF file, 1.7 MB.

## ACKNOWLEDGMENTS

We thank Marta Mollerach (Facultad de Farmacia y Bioquímica, Universidad de Buenos Aires), Juan P. Scapini (Idymic, Rosario, Santa Fe), Luis Calvinho (INTA, Rafaela, Santa Fe), and Gerardo Leotta (Universidad Nacional de La Plata, Buenos Aires) for kindly supplying *S. aureus* isolates. In all cases, the strains were anonymized. CNS strains were generously supplied by M. Alzamura and C. Vay (Universidad de Buenos Aires, Facultad de Farmacia y Bioquímica, Departamento de Bioquímica Clínica, Hospital de Clínicas José de San Martín, Argentina). We thank R. Novick for his kind gift of plasmid pCN53.

C.A.S. is a career researcher of CONICET. C.A.B., N.P.-B., and V.A. were fellows from CONICET during this study. S.T.C. and F.N.A.B. acknowledge fellowships from CONICET. H.R.M. is a Principal Investigator from CIUNR, UNR, Rosario, and acknowledges funds granted by ANPCyT (PICT 2017).

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
