## [Reviewer comments · Microbiology Spectrum]

Microbiology Spectrum

Bioinformatic analysis of a set of fourteen temperate bacteriophages isolated from *Staphylococcus aureus* strains highlights their massive genetic diversity

Cristian Suarez, Soledad Carrasco, Facundo Brandolisio, Virginia Abatangelo, Carina Boncompain, Natalia Peressutti-Bacci, and Hector Morbidoni

Corresponding Author(s): Hector Morbidoni, Facultad de Ciencias Médicas-Universidad Nacional de Rosario

Review Timeline:

Submission Date:	January 30, 2022
Editorial Decision:	April 11, 2022
Revision Received:	June 23, 2022
Accepted:	July 4, 2022

Editor: Daria Van Tyne

Reviewer(s): The reviewers have opted to remain anonymous.

Transaction Report:

DOI: <https://doi.org/10.1128/spectrum.00334-22>

April 11, 2022

Dr. Hector R. Morbidoni
Facultad de Ciencias Médicas-Universidad Nacional de Rosario
Laboratorio de Microbiología Molecular
Santa Fe 3100
Rosario, Santa Fe 2000
Argentina

Re: Spectrum00334-22 (Bioinformatic analysis of a set of fourteen temperate bacteriophages isolated from *Staphylococcus aureus* strains highlights their massive genetic diversity)

Dear Dr. Hector R. Morbidoni:

Thank you for submitting your manuscript to Microbiology Spectrum. Your manuscript was reviewed by two experts, and I would now like you to revise your study in line with their feedback. In addition to the reviewers comments, I found it hard to distinguish some of the colors used in Figure 8. I would suggest you either select more distinct colors, or better yet separate the "S. aureus subcluster" and "other host" annotations into different rings on your iTOL tree.

Link Not Available

Sincerely,

Daria Van Tyne

Journals Department
Reviewer comments:

Reviewer #1 (Comments for the Author):

In the present manuscript the authors have partially characterized in silico 14 temperate bacteriophages infecting *Staphylococcus aureus*. The authors have compared these phages with others already deposited in the public databases. Unfortunately, the work, in its current form, is merely descriptive, and limited to a few prophages. Some experiments, characterizing some of the analyzed modules, would significantly improve the value of the manuscript. For example, the

manuscript highlights the importance of the *S. aureus* phages mobilizing SaPIs. However, this is not tested experimentally. It is also surprising that the impact that these phages could have promoting the transfer of chromosomal genes by lateral transduction, or the transfer of plasmids by generalized transduction, is not even mentioned. Finally, some of the work attributed to Novick was not really performed in his lab (especially that related to the SaPIs). There are other important papers related to the biology and diversity of the *S. aureus* phages that are just ignored.

Reviewer #2 (Comments for the Author):

The manuscript describes a detailed analysis of 14 temperate phages isolated from human and animal sources. The bioinformatic analysis appears well conducted. However, the novelty of the study appears limited, and it should be highlighted which novel or interesting features are to be recognized in these phages or novel knowledge that they provide. The authors cite two recent papers (Moller et al., 2021 (58) and Góller et al., 2021 (62)) that both are large studies of staphylococcal phages. Does the analysis of the present 14 phages provide additional insight into phage biology or importance to the hosting *S. aureus*?

Somewhat puzzling the authors completely failed to detect Sa3int phages even though they are analyzing human strains of *S. aureus*. They propose that this could be due to the strains not being clinical (line 499) however it could be due to the fact that these phages often make small plaques and thus, they are overlooked on a regular basis. I suggest that the authors check their human strains by PCR to determine if Sa3int phages are present. With this a comment could be added about the role of these phage in colonization of healthy individual. In general, the authors should also argue why phages are isolated rather than deduced from genome sequences.

Abstract: The abstract should reflect any key findings or conclusions made rather than "just" convey that an additional 14 phages have been analyzed. Can any conclusions be made when correlating phages with the origin of the staphylococcal strains?

I. 27: Did you expect antibiotic resistance genes encoded in the phages? this is generally not the case

I. 80: 100 ml?

I. 292-295: I do not see that the cited reference supports that the integrases should be identical? Are the phages from the same source - could there have been recombination?

Result section: Possibly the text in the result section could be limited as most parts are well supported by the figures and it is difficult to read all the listings of phages and properties.

Staff Comments:

Preparing Revision Guidelines

Please return the manuscript within 60 days; if you cannot complete the modification within this time period, please contact me. If you do not wish to modify the manuscript and prefer to submit it to another journal, please notify me of your decision immediately so that the manuscript may be formally withdrawn from consideration by Microbiology Spectrum.

If your manuscript is accepted for publication, you will be contacted separately about payment when the proofs are issued; please follow the instructions in that e-mail. Arrangements for payment must be made before your article is published. For a

complete list of **Publication Fees**, including supplemental material costs, please visit our website.

ANSWERS TO REVIEWERS

Dear editor

We thank you and the colleagues who reviewed our manuscript, all the comments have been very valuable for the improvement of our manuscript.

As general comments for both reviewers we have to emphasize that one of the aspects that make this manuscript worth publishing is the geographic location from which strains are obtained. The results may not be as novel as desired, however, even matching data is scientifically useful to validate information and create knowledge. We would like to bring to the attention of the reviewers that our experimental work was conducted on a comparable time frame than publications from other groups cited in our manuscript; however, we were not able to publish our results in a timely manner.

A point that we would like to be taken in consideration is that budget constrains prevent us of performing WGS as a general strategy. Primer ordering also is a long process of up to two-three weeks. We believe that we have addressed both reviewer's concerns and suggestions and we hope that our manuscript would be considered suitable for publication in its revised version.

Figure 8 has been improved as requested by the editor

A new co-author has been added due to his contribution to the final revision

REVIEWER #1

In the present manuscript the authors have partially characterized in silico 14 temperate bacteriophages infecting *Staphylococcus aureus*. The authors have compared these phages with others already deposited in the public databases. Unfortunately, the work, in its current form, is merely descriptive, and limited to a few prophages. Some experiments, characterizing some of the analyzed modules, would significantly improve the value of the manuscript. For example, the manuscript highlights the importance of the *S. aureus* phages mobilizing SaPIs. However, this is not tested experimentally. It is also surprising that the impact that these phages could have promoting the transfer of chromosomal genes by lateral transduction, or the transfer of plasmids by generalized transduction, is not even mentioned. Finally, some of the work attributed to Novick was not really performed in his lab (especially that related to the SaPIs). There are other important papers related to the biology and diversity of the *S. aureus* phages that are just ignored.

REPLY: We do appreciate the reviewer's comments on our manuscript. It should be emphasized that the manuscript title indicates that our work focuses on the bioinformatics analysis which by its nature, is descriptive.

We have no means of obtaining materials to test SaPIs mobilization in a timely manner, however, we tested phage infection on several clinical CNS isolates (results added as Supplementary Figure 2) as well as performed transduction assays using the only *Staphylococcus* plasmid we currently have. We really hope the added work would satisfy the reviewer's concerns.

Credit for the work on SAPIs is shown by the chosen references, lines 48-51 have been rephrased. We apologize for any mistake done.

REVIEWER #2

The manuscript describes a detailed analysis of 14 temperate phages isolated from human and animal sources. The bioinformatic analysis appears well conducted. However, the novelty of the study appears limited, and it should be highlighted which novel or interesting features are to be recognized in these phages or novel knowledge that they provide. The authors cite two recent papers (Moller et al., 2021 (58) and Góller et al., 2021 (62)) that both are large studies of staphylococcal phages. Does the analysis of the present 14 phages provide additional insight into phage biology or importance to the hosting *S. aureus*?

REPLY: Part of the concerns of the reviewer have been addressed by our general comments to both reviewers. We agree that Moller's and Goller's papers are indeed large studies conducted by research groups, those publications came out while we were drafting our manuscript. Of note, we have performed host range studies for selected phages of our set and found that they can infect some *S. hominis* strains, in line with reports by other authors. This supports publications referred to inter-species gene transfer. It is of interest that one of the phages we describe, vB_Sau_308, although not unique, is different in several aspects of gene contents and organization

Somewhat puzzling the authors completely failed to detect Sa3int phages even though they are analyzing human strains of *S. aureus*. They propose that this could be due to the strains not being clinical (line 499) however it could be due to the fact that these phages often make small plaques and thus, they are overlooked on a regular basis. I suggest that the authors check their human strains by PCR to determine if Sa3int phages are present. With this a comment could be added about the role of these phage in colonization of healthy individual. In general, the authors should also argue why phages are isolated rather than deduced from genome sequences.

REPLY: During our initial screening only one lysis plaque was kept from each induced strain; although we believe we were unbiased in our selection of plaques since we looked at each plate under a magnifying glass, there is a possibility that very small sized plaques were overlooked. We believe that the reviewer's suggestion to perform PCR on the original strains is a logical option, however, due to both lack of funds and time constrains for resubmission, we preferred to change the sentence as it does not alter the essence of our results. Although WGS would allow to obtain much more information on the features of clinical strains besides and beyond phage genome identification, it is more expensive with the additional disadvantage of not producing phage particles which in the end, are necessary.

Abstract: The abstract should reflect any key findings or conclusions made rather than "just" convey that an additional 14 phages have been analyzed. Can any conclusions be made when correlating phages with the origin of the staphylococcal strains?

REPLY: Addressed in the text.

I. 27: Did you expect antibiotic resistance genes encoded in the phages? this is generally not the case

The reviewer is right on this point and we are sorry for the misleading statement. We have eliminated the reference to antibiotic resistance genes, focusing on virulence factors instead. On this

topic, we have found that one of our phages does not contain such genes in spite that the rest of the members of the subcluster do.

I. 80: 100 ml? corrected

I. 292-295: I do not see that the cited reference supports that the integrases should be identical? Are the phages from the same source - could there have been recombination?

REPLY: The paragraph has been reorganized as it was misleading. We did detect a complete identity between the S-Int of our phages as well as other phages in our local database. In addition to that the rest of S-Int sequences displayed a very homology too. Thus, we added a phylogenetic tree to show that and modified Supplementary Figure 1 .

Result section: Possibly the text in the result section could be limited as most parts are well supported by the figures and it is difficult to read all the listings of phages and properties.

REPLY: done as requested.

July 4, 2022

Dr. Hector R. Morbidoni
Facultad de Ciencias Médicas-Universidad Nacional de Rosario
Laboratorio de Microbiología Molecular
Santa Fe 3100
Rosario, Santa Fe 2000
Argentina

Re: Spectrum00334-22R1 (Bioinformatic analysis of a set of fourteen temperate bacteriophages isolated from *Staphylococcus aureus* strains highlights their massive genetic diversity)

Dear Dr. Hector R. Morbidoni:

Your manuscript has been accepted, and I am forwarding it to the ASM Journals Department for publication. You will be notified when your proofs are ready to be viewed.

Sincerely,

Daria Van Tyne
Editor, Microbiology Spectrum
